# Beyond Average Return in Markov Decision Processes

**Alexandre Marthe**
UMPA
ENS de Lyon
Lyon, France
alexandre.marthe@ens-lyon.fr

**Aurelien Garivier**
UMPA UMR 5669 and LIP UMR 5668
Univ. Lyon, ENS de Lyon
46 allée d'Italie F-69364 Lyon cedex 07, France
aurelien.garivier@ens-lyon.fr

**Claire Vernade**
University of Tuebingen
Tuebingen, Germany
claire.vernade@uni-tuebingen.de

## Abstract

What are the functionals of the reward that can be computed and optimized exactly in Markov Decision Processes? In the finite-horizon, undiscounted setting, Dynamic Programming (DP) can only handle these operations efficiently for certain classes of statistics. We summarize the characterization of these classes for policy evaluation, and give a new answer for the planning problem. Interestingly, we prove that only generalized means can be optimized exactly, even in the more general framework of Distributional Reinforcement Learning (DistRL). DistRL permits, however, to evaluate other functionals approximately. We provide error bounds on the resulting estimators, and discuss the potential of this approach as well as its limitations. These results contribute to advancing the theory of Markov Decision Processes by examining overall characteristics of the return, and particularly risk-conscious strategies.

## 1 Introduction

Reinforcement Learning (RL) has emerged as a flourishing field of study, delivering significant practical applications ranging from robot control and game solving to drug discovery or hardware design [Lazic et al., 2018, Popova et al., 2018, Volk et al., 2023, Mirhoseini et al., 2020]. The cornerstone of RL is the "return" value, a sum of successive rewards. Conventionally, the focus is on computing and optimizing its expected value on Markov Decision Process (MDP). The remarkable efficiency of MDPs comes from their ability to be solved through dynamic programming with the Bellman equations [Sutton and Barto, 2018, Szepesvári, 2010]. RL theory has seen considerable expansion, with a renewed interest for the consideration of more rich descriptions of a policy's behavior than the sole average return. At the other end of the spectrum, the so-called *Distributional Reinforcement Learning* (DistRL) approach aims at studying and optimizing the entire return distribution, leading to impressive practical results [Bellemare et al., 2017, Hessel et al., 2018, Wurman et al., 2022, Fawzi et al., 2022]. Between the expectation and the entire distribution, the efficient handling of other statistical functionals of the reward appears also particularly relevant for risk-sensitive contexts [Bernhard et al., 2019, Mowbray et al., 2022].

Despite recent progress, the full understanding of the abilities and limitations of DistRL to compute other functionals remains incomplete, with the underlying theory yet to be fully understood. Historically, the theory of RL has been established for discounted MDPs, see e.g. [Sutton and Barto, 2018, Watkins and Dayan, 1992, Szepesvári, 2010, Bellemare et al., 2023] for modern reference textbooks. Recently more attention was drawn to the undiscounted, finite-horizon setting [Auer, 2002,

37th Conference on Neural Information Processing Systems (NeurIPS 2023).

Osband et al., 2013, Jin et al., 2018, Ghavamzadeh et al., 2020], for which fundamental questions remain open. In this paper, we explore policy evaluation, planning and exact learning algorithms for undiscounted MDPs for the optimization problem of general functionals of the reward. We explicitly delimit the possibilities offered by dynamic programming as well as DistRL.

Our paper specifically addresses two questions:

  (i) How accurately can we evaluate statistical functionals by using DistRL?
  (ii) Which functionals can be exactly optimized through dynamic programming?

We first recall the fundamental results in dynamic programming and Distributional RL. Addressing question (i), we refer to Rowland et al. [2019]'s results on Bellman closedness and provide their adaptation to undiscounted MDPs. We then prove upper bounds on the approximation error of Policy Evaluation with DistRL and corroborate these bounds with practical experiments. For question (ii), we draw a connection between Bellman closedness and planning. We then utilize the DistRL framework to identify two key properties held by *optimizable* functionals.

Our main contribution is a characterization of the families of utilities that verify these two properties (Theorem 2). This result gives a comprehensive answer to question (ii) and closes an important open issue in the theory of MDP. It shows in particular that DistRL does not extend the class of functionals for which planing is possible beyond what is already allowed by classical dynamic programming.

## 2 Background

We introduce the classical RL framework in finite-horizon tabular Markov Decisions Processes (MDPs). We write $\mathscr{P}(\mathbb{R})$ the space of probability distributions on $\mathbb{R}$. A finite-horizon tabular MDP is a tuple $\mathcal{M} = (\mathcal{X}, \mathcal{A}, p, R, H)$, where $\mathcal{X}$ is a finite state space, $\mathcal{A}$ is a finite action space, $H$ is the horizon, for each $h \in [H]$, $p_h(x, a, \cdot)$ is a transition probability law and $R_h(x, a)$ is a reward random variable with distribution $\varrho_h$. The parameters $(p_h)$ and $(R_h)$ define the *model* of dynamics. A deterministic policy on $\mathcal{M}$ is a sequence $\pi = (\pi_1, \ldots, \pi_H)$ of functions $\pi_h : \mathcal{X} \to \mathcal{A}$.

Reinforcement Learning traditionally focuses on learning policies optimizing the expected return. For a given policy $\pi$, the $Q$-function maps a state-action pair to its expected return under $\pi$:

$$Q_h^\pi(x, a) = \mathbb{E}_{\varrho_h}[R_h(x, a)] + \sum_{s'} p_h(x, a, x')Q_{h+1}^\pi(x', \pi_{h+1}(x')), \qquad Q_{H+1}^\pi(x, a) = 0 . \quad (1)$$

When the model is known, the $Q$-function of a policy $\pi$ can be computed by doing a backward recursion, also called dynamic programming. This is referred to as *Policy Evaluation*. Similarly, an optimal policy can be found by solving the optimal Bellman equation:

$$Q_h^*(x, a) = \mathbb{E}_{\varrho_h}[R_h(x, a)] + \sum_{x'} p_h(x, a, x') \max_{a'} Q_{h+1}^*(x', a'), \qquad Q_{H+1}^*(x, a) = 0 . \quad (2)$$

Solving this equation when the model is known is also called *Planning*. When it is unknown, *reinforcement learning* aims at finding the optimal policy from sample runs of the MDP. But evaluating and optimizing the *expectation* of the return in the definition of the Q-function above is just one choice of statistical functional. We now introduce Distributional RL and then discuss other statistical functionals that generalize the expected setting discussed so far.

### 2.1 Distributional RL

Distributional RL (DistRL) refers to the approach that tracks not just a statistic of the return for each state but its *entire distribution*. We introduce here the most important basic concepts and refer the reader to the recent comprehensive survey by Bellemare et al. [2023] for more details. The main idea is to use the full distributions to estimate and optimize various metrics over the returns ranging from the mere expectation [Bellemare et al., 2017] to more complex metrics [Rowland et al., 2019, Dabney et al., 2018a, Liang and Luo, 2022].

At state-action $(x, a)$, let $Z_h^\pi(x, a)$ denote the future sum of rewards when following policy $\pi$ and starting at step $h$, also called *return*. It verifies the simple recursive formula $Z_h^\pi(x, a) = R_h(x, a) + Z_{h+1}^\pi(X', \pi_{h+1}(X'))$ where $X' \sim p_{h+1}(x, a, \cdot)$. Its distribution is $\eta = (\eta_h^\pi(x, a))_{(x, a, h) \in \mathcal{X} \times \mathcal{A} \times [H]}$ and is often referred to as the *Q-value distribution*. One can easily derive the recursive law of the return as a convolution: for any two measures $\nu_1, \nu_2 \in \mathscr{P}(\mathbb{R})$, we denote their convolution by $\nu_1 * \nu_2(t) = \int_\mathbb{R} \nu_1(\tau)\nu_2(t - \tau)d\tau$. For any two independent random variables $X$ and $Y$, the distribution of the sum $Z = X + Y$ is the convolution of their distributions: $\nu_Z = \nu_X * \nu_Y$. Thus, the law of $Z_h^\pi(x, a)$ is

$$\forall x, a, h, \quad \eta_h^\pi(x, a) = \varrho_h(x, a) * \sum_{x'} p_h(x, a, x')\eta_h^\pi(x', \pi_{h+1}(x')) . \tag{3}$$

This equation is a distributional equivalent to Eq. (1) and thus defines a *distributional Bellman operator* $\eta_h^\pi = \mathcal{T}_h^\pi \eta_{h+1}^\pi$.

Obviously, from a practical point of view, distributions form a non-parametric family that is not computationally tractable. It is necessary to choose a parametric (thus incomplete) family to represent them. Even the restriction to discrete reward distributions is not tractable, since the number of atoms in the distributions may grow exponentially with the number of steps[1] [Achab and Neu, 2021]: approximations are unavoidable. The most natural solution is to use projections of the obtained distribution on the parametric family, at each step of the Bellman operator. This process is called *parameterization*. The practical equivalent to Eq. (1) in DistRL hence writes

$$\forall x, a, h, \quad \hat{\eta}_h^\pi(x, a) = \Pi\left(\varrho_h(x, a) * \sum_{x'} p_h(x, a, x')\hat{\eta}_{h+1}^\pi(x', \pi_{h+1}(x'))\right) , \tag{4}$$

where $\Pi$ is the projector operator on the parametric family. The full policy evaluation algorithm in DistRL is summarized in Alg.1.

---

**Algorithm 1** Policy Evaluation (Dynamic Programming) for Distributional RL

---

**Input:** model $p$, reward distributions $\varrho_h$, policy $\pi$ to evaluated, $\Pi$ projection.
**Data:** $\eta \in \mathbb{R}^{H|\mathcal{X}||\mathcal{A}|N}$
$\forall x, a \in \mathcal{X} \times \mathcal{A}, \quad \eta_H(x, a) = \delta_0$
**for** $h = H - 1 \to 0$ **do**
$\quad \eta_h(x, a) = \varrho_h(x, a) * \sum_{x'} p_h(x, a, x')\eta_{h+1}(x', \pi_{h+1}(x')) \quad \forall x, a \in \mathcal{X} \times \mathcal{A}$
$\quad \eta_h(x, a) = \Pi(\eta_h(x, a)) \quad \forall x, a \in \mathcal{X} \times \mathcal{A}$
**end for**
**Output:** $\eta_h(x, a) \forall x, a, h$

---

**Distribution Parametrization** The most commonly used parametrization is the so-called *quantile projection*. It put Diracs (atoms) with fixed weights at locations that correspond to the quantiles of the source distribution. One main benefit is that it does not require a previous knowledge of the support of the distribution, and allows for unbounded distributions.

The quantile projection is defined as

$$\Pi_Q\mu = \frac{1}{N}\sum_{i=0}^{N-1}\delta_{z_i}, \quad \text{with } (z_i)_i \text{ chosen such as } F_\mu(z_i) = \frac{2i + 1}{2N} , \tag{5}$$

which corresponds to a minimal $W_1$ distance: $\Pi_Q\mu \in \arg\min_{\hat{\mu} = \sum_i \delta_i/N} W_1(\mu, \hat{\mu})$, where $W_1(.,.)$ is the Wasserstein distance defined for any distributions $\nu_1, \nu_2$ as $W_1(\nu_1, \nu_2) = \int_0^1 \left|F_{\nu_1}^{-1}(u) - F_{\nu_2}^{-1}(u)\right| \, du$. Note that this parametrization might admit several solutions and thus the projection may not be unique. For simplicity, we overload the notation to $\Pi_Q\eta = (\Pi_Q\eta(x, a))_{(x, a) \in \mathcal{X} \times \mathcal{A}}$

---

[1]The support of the return (sum of the rewards) is incremented at each step by a number of atoms that depend on the current support.

For a Q-value distribution $\eta$ with support of length $\Delta_\eta$, and parametrization of resolution $N$, Rowland et al. [2019] prove that the projection error is bounded by

$$\sup_{(x,a)\in\mathcal{X}\times\mathcal{A}} W_1(\Pi_Q\eta(x,a),\eta(x,a)) \leq \frac{\Delta_\eta}{2N} \; . \tag{6}$$

In Section 3, we extend this result to the full iterative Policy Evaluation process and bound the error on the returned statistical functional in the finite-horizon setting. Note that other studied parametrizations exist but are less practical. For completeness, we discuss the Categorical Projection [Bellemare et al., 2017][Rowland et al., 2018][Bellemare et al., 2023] in Appendix B.

## 2.2 Beyond expected returns

The expected value is an important functional of a probability distribution, but it is not the only one of interest in decision theory – especially when a control of the risk is important. We discuss two concepts that have received considerable attention: *utilities*, defined as expected values of functions of the return, and *distorted means* which place emphasis on certain quantiles.

**Expected Utilities** are of the form $\mathbb{E}[f(Z)]$, or $\int f \, d\nu$, where $Z$ is the return of distribution $\nu$ and $f$ is an increasing function. For instance, when $f$ is a power function, we obtain the different moments of the return. The case of exponential functions plays a particularly important role: the resulting utility is referred to as *exponential utility*, *exponential risk measure*, or *generalized mean* according to the context:

$$U_{\exp}(\nu) = \frac{1}{\lambda} \log \mathbb{E}\big[\exp(\lambda X)\big] \quad X \sim \nu \text{ and } \lambda \in \mathbb{R} \; . \tag{7}$$

This family of utilities has a variety of applications in finance, economics, and decision making under uncertainty [Föllmer and Schied, 2016]. They can be considered as a risk-aware generalization of the expectation, with benefits such as accommodating a wide range of behaviors [Shen et al., 2014] from risk-seeking when $\lambda > 0$, to risk-averse when $\lambda < 0$ (the limit $\lambda \to 0$ is exactly the expectation). To fix ideas, $U_{\exp}\big(\mathcal{N}(\mu,\sigma^2)\big) = \mu + \lambda\sigma^2$: each $\lambda$ captures a certain quantile of the Gaussian distribution.

**Distorted means**, on the other hand, involve taking the mean of a random variable, but with a different weighting scheme [Dabney et al., 2018a]. The goal is to place more emphasis on certain quantiles, which can be achieved by considering the quantile function $F^{-1}$ of the random variable and a continuous increasing function $\beta : [0,1] \to [0,1]$. By applying $\beta$ to a uniform variable $\tau$ on $[0,1]$ and evaluating $F^{-1}$ at the resulting value $\beta(\tau)$, we obtain a new random variable that takes the same values as the original variable, but with different probabilities. The distorted mean is then calculated as the mean of this new random variable, given by the formula $\int \beta'(\tau)F^{-1}(\tau)d\tau$. If $\beta$ is the identity function, the result is the classical mean. When $\beta$ is $\tau \mapsto \min(\tau/\alpha,1)$, we get the $\alpha$-Conditional Value at Risk (CVaR($\alpha$)) of the return, a risk measure widely used in risk evaluation [Rockafellar et al., 2000].

## 3 Policy Evaluation

The theory of MDPs is particularly developed for estimating and optimizing the mean of the return of a policy. But other values associated to the return can be computed the same way, by dynamic programming. This includes for instance the variance of the return, or more generally, any moment of order $p \geq 2$, as was already noticed in the 1980's [Sobel, 1982]. Recently, Rowland et al. [2019] showed that for utilities in discounted MDPs, this is essentially all that can be done. More precisely, they introduce the notion of *Bellman closedness* (recalled below for completeness) that characterizes a finite set of statistics that can efficiently be computed by dynamic programming.

**Definition 1** (Bellman closedness [Rowland et al., 2019]). *A set of statistical functionals $\{s_1, \ldots s_K\}$ is said to be* Bellman closed *if for each $(x,a) \in \mathcal{X} \times \mathcal{A}$, the statistics $s_{1:K}(\eta_h^\pi(x,a))$ can be expressed in closed form in terms of the random variables $R_h(x,a)$ and $(s_{1:K}(\eta_{h+1}^\pi(X',A')),\ A' \sim \pi(x),\ X' \sim p_h(x,A',\cdot)$, independently of the MDP.*

Importantly, in the undiscounted setting, Rowland et al. [2019](Appendix B, Theorem 4.3) show that the only families of utilities that are Bellman closed are of the form $\{x \mapsto x^\ell \exp(\lambda x) | 0 \leq \ell \leq L\}$

for some $L < \infty$. Thus, all utilities and statistics of the form of (or linear combinations of) moments and exponential utilities can easily be computed by classic linear dynamic programming and do not require distributional RL (see Appendix A.3).

Some important metrics such as the CVaR or the quantiles are not known to belong to any Bellman-closed set and hence cannot be easily computed. For this kind of function of the return, the knowledge of the transitions and the values in following steps is insufficient to compute the value on a specific step. In general, it requires the knowledge of the whole distribution of each reward in each state. Hence, techniques developed in distributional RL come in handy: for a choice of parametrization, one can use the projected dynamic programming step Eq. (4) to propagate a finite set of values along the MDP and approximate the distribution of the return. In the episodic setting, following the line of Rowland et al. [2019] (see Eq.(6)), we prove that the Wasserstein distance error between the exact and approximate distribution of the Q-values of a policy is bounded.

**Proposition 1.** *Let $\pi$ be a policy and $\eta^\pi$ the associated Q-value distributions. Assume the return is bounded on a interval of length $\Delta_\eta \le H\Delta_R$, where $\Delta_R$ is the support size of the reward distribution. Let $\hat{\eta}^\pi$ be the Q-value distributions obtained by dynamic programming (Algorithm 1) using the quantile projection $\Pi_Q$ with resolution $N$. Then,*

$$\sup_{(x,a,h) \in (\mathcal{X}, \mathcal{A}, [H])} W_1(\hat{\eta}_h^\pi(x,a), \eta_h^\pi(x,a)) \le H\frac{\Delta_\eta}{2N} \le H^2\frac{\Delta_R}{2N} \ .$$

This result shows that the loss of information due to the parametrization may only grow quadraticly with the horizon. The proof consists of summing the projection bound in (6) at each projection step, and using the non-expansion property of the Bellman operator [Bellemare et al., 2017]. The details can be found in Appendix C

The key question is then to understand how such error translates into our estimation problem when we apply the function of interest to the approximate distribution. We provide a first bound on this error for the family of statistics that are either utilities or distorted means.

First, we prove that the utility is Lipschitz on the set of return distributions.

**Lemma 1.** *Let $s$ be either an utility or a distorted mean and let $L$ be the Lipschitz coefficient of its characteristic function. Let $\nu_1, \nu_2$ be return distributions. Then:*

$$|s(\nu_1) - s(\nu_2)| \le LW_1(\nu_1, \nu_2) \ .$$

Both family of functionals are treated separately, but lays a similar bound. The utility bound is the direct application of the Kantorovitch-Rubenstein duality, while the distorted mean one is a direct majoration in the integral. Again, the details are provided in the Appendix.

This property allows us to prove a maximal upper bound on the estimation error for those two families.

**Theorem 1.** *Let $\pi$ be a policy. Let $\eta^\pi$ be the Q-value return distribution associated to $\pi$ with the return bounded on a interval of length $\Delta_\eta \le H\Delta_R$ where $\Delta_R$ is the support size of the reward distribution. Let $\hat{\eta}^\pi$ be the approximated return distribution computed with Algorithm 1, for the projection $\Pi_Q$ with resolution $N$. Let $s$ be either an expected utility or a distorted mean, and $L$ the Lipschitz coefficient of its characteristic function. Then:*

$$\sup_{x,a,h} |s(\hat{\eta}_h^\pi(x,a)) - s(\hat{\eta}_h^\pi(x,a))| \le LH\frac{\Delta_\eta}{2N} \le LH^2\frac{\Delta_R}{2N} \ .$$

Note that depending on the choice of utilities, the Lipschitz coefficient $L$ may also depend on $H$ and $\Delta R$. For instance, in a stationary MDP, the Lipschitz constant of the exponential utility depends exponentially on $\Delta_\eta$. For the CVaR($\alpha$), however, $L$ is constant and only depends on $\alpha \in (0, 1)$.

**Experiment: empirical validation of the bounds on a simple MDP**  We consider a simple Chain MDP environment of length $H = 70$ equal to the horizon (see Figure 1 (right)) [Rowland et al., 2019], with a single action leading to the same discrete reward distribution for every step. We consider a Bernouilli reward distribution $\mathcal{B}(0.5)$ for each state so that the number of atoms for the return only grows linearly[2] with the number of steps, which allows to compute the exact distribution easily.

---

[2]At round $h \in [H]$, the support of the return is $\{0, 1, ..., h\}$, so $h$ atoms.

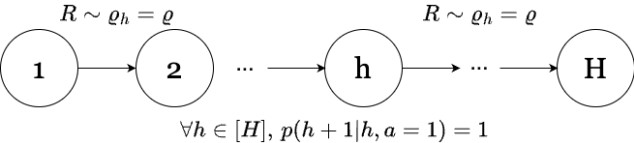

$$\forall h \in [H], \ p(h+1|h, a=1) = 1$$

Figure 1: A Chain MDP of length $H$ with deterministic transition and identical reward distribution for each state.

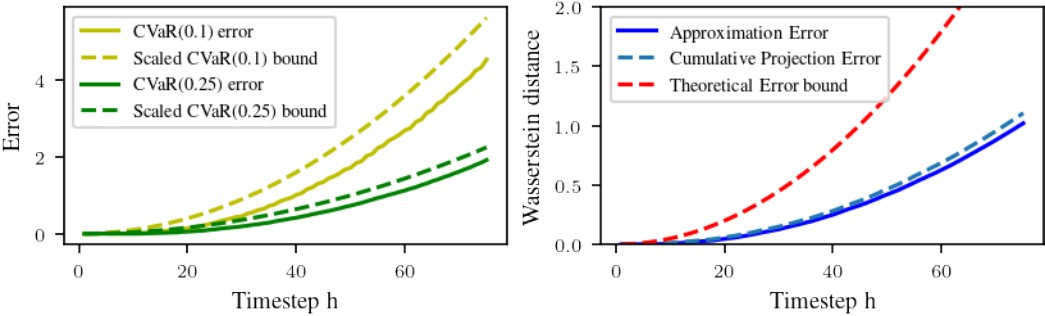

Figure 2: Left: Validation of Theorem 1 on $CVaR(\alpha)$ together with the scaled upper bound (see main text for discussion): the quadratic dependence in $H$ is verified. Right: Validation of Proposition 1: The cumulative projection error (dashed blue) is the sum of the projection errors at every time step, and matches the true approximation error (solid blue). The theoretical upper bound (dashed red) differs only by a factor 2.

We compare the distributions obtained with exact dynamic programming and the approximate distribution obtained by Alg 1, with a quantile projection with resolution $N = 1000$. Note that even at early stages, when the true distribution has less atoms than the resolution, the exact and approximate distributions differ due to the weights of the atoms in the quantile projection. Figure 2 (Right) reports the Wasserstein distance between the two distributions: the cumulative projection approximation error (dashed blue), the true error between the current exact and approximate distributions (solid blue) and the theoretical bound (red). Fundamentally, the proof of Prop. 1 upper bounds the distance between distributions by the cumulative projection error so we plot this quantity to help validating it.

We also empirically validate Theorem 1 by computing the $CVaR(\alpha)$ for $\alpha \in \{0.1, 0.25\}$, corresponding respectively to distorted means with Lipschitz constants $L = \{10, 4\}$. We compute these statistics for both distributions and report the maximal error together with the theoretical bound, re-scaled[3] by a factor 5. Figure 2 (Left) shows an impressive correspondence of the theory and the empirical results despite a constant multiplicative gap.

## 4   Planning

Planning refers to the problem of returning a policy that optimizes our objective for a given model and reward function (or distribution in DistRL). It shares with policy evaluation the property to be grounded on a Bellman equation: see Eq. (2) for the classical expected return, which leads to efficient computations by dynamic programming.

For other statistical functionals of the cumulated reward, however, can the optimal policy be computed efficiently? The main result of this section characterizes the family of functionals that can be exactly and efficiently optimized by Dynamic Programming.

In the previous section, we recalled that Bellman-closed sets of utilities can be efficiently computed by DP as long as all the values of the utilities in the Bellman-closed family are computed together. For the planning problem, however, we only want to optimize one utility so we cannot consider families as previously. Under this constraint, only exponential and linear expected utilities are Bellman closed and thus can verify a Bellman Equation. In fact, for the exponential utilities, such Bellman Equation

---

[3]Scaling by a constant factor allows us to show the corresponding quadratic trends.

exists and allows for the planning problem to be solved efficiently [Howard and Matheson, 1972]:

$$Q_h^\lambda(x, a) = U_{\exp}^\lambda(R_h(x, a)) + \frac{1}{\lambda} \log \left[ \sum_{x'} p_h(x, a, x') \exp \left( \lambda \max_{a'} Q_{h+1}^\lambda(x', a') \right) \right] \quad (8)$$

$$\text{with } Q_{H+1}^\lambda(x, a) = 0 \; .$$

However, the question of the existence of Optimal Bellman Equations for non-utility functionals remains open (e.g. quantiles). More generally, an efficient planning strategy is not known. To address these questions, we consider the most general framework, DistRL, and recall the theoretical DP equations for any statistical functional $s$ in the Pseudo[4]-Algorithm 2. DistRL offers the most comprehensive, or 'lossless', approach, so if a statistical functional cannot be optimized with Alg. 2, then there cannot exist a Bellman Operator to perform exact planning.

---

**Algorithm 2** Pseudo-Algorithm: Exact Planning with Distributional RL

---

1: **Input:** model $p$, reward $R$, statistical functional $s$
2: **Data:** $\eta \in \mathbb{R}^{H|\mathcal{X}||\mathcal{A}|N}, \nu \in \mathbb{R}^{H|\mathcal{X}|N}$
3: $\forall x \in \mathcal{X}, \quad \nu_{H+1}^x = \delta_0$
4: **for** $h = H \to 1$ **do**
5: $\quad \eta_h(x, a) = \varrho_h^{(x,a)} * \sum_{x'} p_h^a(x, x')\nu_{h+1}^{x'} \quad \forall x, a \in \mathcal{X} \times \mathcal{A}$
6: $\quad \nu_h^x = \eta_h(x, a^*), \quad a^* \in \arg\max_a s(\eta_h(x, a)) \quad \forall x \in \mathcal{X}$
7: **end for**
8: **Output:** $\eta_h(x, a) \; \forall x, a, h$

---

We formalize this idea with the new concept of *Bellman Optimizable* statistical functionals:

**Definition 2** (Bellman Optimizable statistical functional). *A statistical functional $s$ is called* Bellman optimizable *if, for any MDP $\mathcal{M}$, the Pseudo-Algorithm 2 outputs an optimal return distribution $\eta = \eta^*$ that verifies:*

$$\forall x, a, h, \quad s(\eta_h^*(x, a)) = \sup_\pi s(\eta_h^\pi(x, a)) \; . \quad (9)$$

**Remark.** *This definition is equivalent to the satisfaction of an Optimal Distributionnal Bellman equation. Indeed, a statistical functional $s$ is Bellman Optimizable if and only if, for any $(\mathcal{X}, \mathcal{A}, \varrho, p, \eta)$, $s$ verifies*

$$\sup_{a_{x'}} s \left( \varrho * \sum_{x'} p(x')\eta(x', a_{x'}) \right) = s \left( \varrho * \sum_{x'} p(x')\eta(x', a_{x'}^*) \right)$$

*with $a_x^\star \in \arg\max_a s(\eta(x, a))$*

We can now state our main results that characterizes all the *Bellman optimizable* statistical functionals. First, we prove that such a functional must satisfy two important properties.

**Lemma 2.** *A Bellman optimizable functional $s$ satifies the two following properties:*

- *Independence Property: If $\nu_1, \nu_2 \in \mathscr{P}(\mathbb{R})$ are such that $s(\nu_1) \geq s(\nu_2)$, then*

$$\forall \nu_3 \in \mathscr{P}(\mathbb{R}), \forall \lambda \in [0, 1], \quad s(\lambda\nu_1 + (1-\lambda)\nu_3) \geq s(\lambda\nu_2 + (1-\lambda)\nu_3) \; .$$

- *Translation Property: Let $\tau_c$ denote the translation on the set of distributions: $\tau_c\delta_x = \delta_{x+c}$. If $\nu_1, \nu_2 \in \mathscr{P}(\mathbb{R})$ are such that $s(\nu_1) \geq s(\nu_2)$, then*

$$\forall c \in \mathbb{R}, \quad s(\tau_c\nu_1) \geq s(\tau_c\nu_2) \; .$$

Indeed, the expectation and the exponential utility both satisfy these properties (see Appendix A.2). Each property is implied by an aspect of the Distributional Bellman Equation (Alg. 2, line 5) and the proof (in Appendix E) unveils these important consequences of the recursion identities.

---

[4]This theoretical algorithm handles the full distribution of the return at each step, which cannot be done in practice.

Fundamentally, they follow from the Markovian nature of policies optimized this way: the choice of the action in each state should be independent of other states and rely only on the knowledge of the next-state value distribution.

The Independence property states that, for Bellman optimizable functionals, the value of each next state should not depend on that of any other value in the convex combination in the rightmost term of the convolution. In turn, the Translation property is associated to the leftmost term, the reward, and it imposes that, for Bellman optimizable functionals, the decision on the best action is independent of the previously accumulated reward.

The Independence property is related to expected utilities [von Neumann et al., 1944]. Any expected utility verifies this property (Appendix A.2) but, most importantly, the Expected Utility Theorem (also known as the Von Neumann Morgenstein theorem) implies that any continuous statistical functional $s$ verifying the Independence property can be reduced to an expected utility. This means that for any such statistical functional $s$, there exists $f$ continuous such that $\forall \nu_1, \nu_2 \in \mathscr{P}(\mathbb{R})$, we have $s(\nu_1) > s(\nu_2) \iff U_f(\nu_1) > U_f(\nu_2)$ [von Neumann et al., 1944, Grandmont, 1972].

This result directly narrows down the family of Bellman optimizable functionals to utilities. Indeed, although other functionals might potentially be optimized using the Bellman equation, addressing the problem on utilities is adequate to characterize all possible behaviors. For instance, moment-optimal policies that can be found through dynamic programming, can also be found by optimizing an exponential utility function. The next task is therefore to identify all the utilities that satisfy the second property. We demonstrate that, apart from the mean and exponential utilities, no other $W_1$-continuous functional satisfies this property.

**Theorem 2.** *Let $\varrho$ be a return distribution. The only $W_1$-continuous Bellman Optimizable statistical functionals of the cumulated return are exponential utilities $U_{\exp}(\varrho) = \frac{1}{\lambda} \log \mathbb{E}_\varrho \left[ \exp(\lambda R) \right]$ for $\lambda \in \mathbb{R}$, with the special case of the expectation $\mathbb{E}_\varrho \left[ R \right]$ when $\lambda = 0$.*

Of course, if $U(\rho)$ is a utility and $\psi$ is a monotonous scalar mapping, $\psi(U(\rho))$ is an equivalent utility: one should understand in the previous theorem that a $W_1$-*continuous* Bellman Optimizable statistical functional is equivalent to $U_{\exp}(\varrho)$ for some $\lambda \in \mathbb{R}$. We chose here to define $U_{\exp}(\varrho) = \frac{1}{\lambda} \log \mathbb{E}_\varrho \left[ \exp(\lambda R) \right]$ with the $\log$ and the factor $1/\lambda$ since it results in a normalized utility that tends to the expectation when $\lambda$ goes to $0$. The full proof of Theorem 2 is provided in Appendix E.

We make a few important observations. First, this result shows that algorithms using Bellman updates to optimize any continuous functionals other than the exponential utility cannot guarantee optimality. The theorem does not apply to non-continuous functionals, but Lemma 2 still does. For instance, the quantiles are not $W_1$-continuous so Theorem 2 does not apply, but it is easy to prove that they do not verify the Independence Property and thus are not Bellman Optimizable. Also, there might also exist other optimizable functionals, like moments, but they must first be reduced to exponential or linear utilities.

Most importantly, while in theory, DistRL provides the most general framework for optimizing policies via dynamic programming, our result shows that in fact, the only utilities that can be exactly and efficiently optimized do not require to resort to DistRL. This certainly does not question the very purpose of DistRL, which has been shown to play important roles in practice to regularize or stabilize policies and to perform deeper exploration [Bellemare et al., 2017, Hessel et al., 2018]. Some advantages of learning the distribution lies in the enhanced *robustness* offered in the richer information learned [Rowland et al., 2023], particularly when utilizing neural networks for function approximation [Dabney et al., 2018b, Barth-Maron et al., 2018, Lyle et al., 2019].

## 5 Q-Learning Exponential Utilities

The previous sections consider the the model, i.e. the reward and transition functions, be known. Yet in most practical situations, those are either approximated or learnt[5]. After addressing policy evaluation (Section 3) and planning (Section 4), we conclude here the argument of this paper by addressing the question of learning the statistical functionals of Theorem 2. In fact, we simply highlight a lesser known version of Q-Learning Watkins and Dayan [1992] that extend this celebrated algorithm to exponential utilities. We provide the pseudo-code for the algorithm proposed by Borkar

---

[5]Either explicitly (model-based RL) or implicitly (model-free RL, considered here).

---

**Algorithm 3** Q-Learning for Linear and Exponential Utilities

---
1: **Input:** $(\alpha_t)_{t \in \mathbb{N}}$, transition and reward generator. $Q_h(x, a) \leftarrow H, \forall (x, a, h) \in \mathcal{X} \times \mathcal{A} \times [H]$
2: **Utilities:** Linear $(Z \mapsto \lambda Z + b)$ or Exponential $(Z \mapsto \log(\mathbb{E} \exp(\lambda Z))/\lambda)$
3: **for** episode $K = 1, \ldots, K$ **do**
4:     Observe $x_1 \in \mathcal{X}$
5:     **for** step $h = 1, \ldots, H$ **do**
6:         Choose action $a_h \in \arg \max_{a \in \mathcal{A}} Q_h(x_h, a)$
7:         Observe reward $r_h$ and transition $x_{h+1}$ and update for chosen objective:
8:         Linear Util.: $Q_h(x_h, a_h) \leftarrow (1 - \alpha_k) Q_h(x_h, a_h) + \alpha_k [\lambda(r_h + \max_{a'} Q_{h+1}(x_{h+1}, a')) + b]$
9:         Exponential Util.: $Q_h(x_h, a_h) \leftarrow \frac{1}{\lambda} \log \left[ (1 - \alpha_k) e^{\lambda \cdot Q_h(x_h, a_h)} + \alpha_k e^{\lambda [r_h + \max_{a'} Q_h(x_{h+1}, a')]} \right]$
10:     **end for**
11: **end for**
12: **Output:** $Q_h(x, a) \forall x, a$

---

[2002, 2010] with the relevant utility-based updates in Alg. 3. We refer to these seminal works for convergence proofs. Linear utility updates (line 8) differ only slightly from classical ones for expected return optimization, which have been shown to lead to the optimal value asymptotically [Watkins and Dayan, 1992].

## 6 Discussions and Related Work

**The Discounted Framework** We focused in this article on undiscounted MDPs, and it is important to note that the results differ for discounted scenarios. The crucial difference is that the family of exponential utilities no longer retains Bellman Closed or Bellman Optimizable properties due to the introduction of the discount factor $\gamma$ [Rowland et al., 2019]. When it comes to Bellman Optimization, the necessary translation property becomes an affine property : $\forall c, \gamma, \; s(\tau_c^\gamma \nu_1) > s(\tau_c^\gamma \nu_2)$ where $\tau_c^\gamma$ is the affine operator such that $\tau_c^\gamma \delta_x = \delta_{\gamma x + c}$. This property is not upheld by the exponential utility. Nonetheless, there exists a method to optimize the exponential utility through dynamic programming in discounted MDPs [Chung and Sobel, 1987]. This approach requires modifying the functional to optimize at each step (the step $h$ is optimized with the utility $x \mapsto \exp(\gamma^{-h} \lambda x)$), but it also implies a loss of policy stationarity, property usually obtained in dynamic programming for discounted finite-horizon MDPs [Sutton and Barto, 2018].

**Utilizing functionals to optimize expected return.** DistRL has also been used in Deep Reinforcement learning to optimize non-Bellman-optimizable functionals such as distorted means[Ma et al., 2020, Dabney et al., 2018a]. While, as we proved so, such algorithms cannot lead to optimal policies in terms of these functionals, experiments show that in some contexts they can lead to better expected return and faster convergence in practice. The change of functional can be interpreted as a change in the exploration process, and the resulting risk-sensitive behaviors seem to be relevant in adequate environments.

**Dynamic programming for the optimization of other functionals** To optimize other statistical functionals such as CVaR and other utilities such as moments with Dynamic Programming, Bäuerle and Ott [2011] and Bäuerle and Rieder [2014] propose to extend the state space of the original MDP to $\mathcal{X}' = \mathcal{X} \times \mathbb{R}$ by theoretically adding a continuous dimension to store the current cumulative rewards. This idea does not contradict our results, and the resulting algorithms remain empirically much more expensive.

Another recent thread of ideas to optimize functionals of the reward revolve around the dual formulation of RL through the empirical state distribution [Hazan et al., 2019]. Algorithms can be derived by noticing that utilities like the CVar are equivalent to solving a convex RL problem [Mutti et al., 2023].

## 7 Conclusion

Our work closes an important open problem in the theory of MDPs: we exactly characterize the families of statistical functionals that can be evaluated and optimized by dynamic programming.

We also put into perspective the DistRL framework: the only functionals of the return that can be optimized with DistRL can actually be handled exactly by dynamic programming. Its benefit lies elsewhere, and notably in the improved stability of behavioral properties it allows. We believe that, by narrowing down the avenues to explain its empirical successes, our work can contribute to clarify the further research to conduct on the theory of DistRL.

## Acknowledgements

Alexandre Marthe and Aurélien Garivier acknowledge the support of the Project IDEXLYON of the University of Lyon, in the framework of the Programme Investissements d'Avenir (ANR-16-IDEX-0005), Chaire SeqALO (ANR-20-CHIA-0020-01), and Project FOUNDRY in PEPR-IA.
Claire Vernade is funded by the Deutsche Forschungsgemeinschaft (DFG) under both the project 468806714 of the Emmy Noether Programme and under Germany's Excellence Strategy – EXC number 2064/1 – Project number 390727645. Claire Vernade also thanks the international Max Planck Research School for Intelligent Systems (IMPRS-IS) and Seek.AI for their support.

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

# A   Additional remarks

The Wasserstein metric is defined as $W_1(\nu_1, \nu_2) = \int_0^1 \left| F_{\nu_1}^{-1}(u) - F_{\nu_2}^{-1}(u) \right| \, du$ and the Cramer metric as $\ell_2(\nu_1, \nu_2) = \left( \int_{-\infty}^{+\infty} |F_{\nu_1}(u) - F_{\nu_2}(u)|^2 \, du \right)^{\frac{1}{2}}$. For both metrics, we define their supremum $\overline{\ell_2}(\eta_1, \eta_2) = \sup_{(x,a) \in \mathcal{X} \times \mathcal{A}} \ell_2(\eta_1(x,a), \eta_2(x,a))$ and $\overline{W}_1(\eta_1, \eta_2) = \sup_{(x,a) \in \mathcal{X} \times \mathcal{A}} W_1(\eta_1(x,a), \eta_2(x,a))$.

## A.1   Remarks on the recursive definition of the Q-value distribution

The notation of the Q-value distribution $\eta$ is often deceivingly complex compared to the actual object it means to represent. While the 'usual' expected Q-function $Q(x, a)$ is simply understood as the expected return of a policy at state-action pair $(x, a)$, DistRL requires us to keep a notation for the complete distribution of the return. In other words, the Q-value distribution $\eta_{\pi,h}^{(x,a)}$ should be understood as the distribution of the *random variable* $Z = R + Z(S')$, which is the convolution of the individual distributions of these two independent random variable. It can also be written:

$$\forall x, a, h, \quad \eta_h^\pi(x, a) = \sum_{x', a'} \int p_h(x, a, x') \pi_{h+1}^{x'}(a') \eta_{h+1}^\pi(x', a')(\cdot - r) R_h^{(x,a)}(dr) . \qquad (10)$$

## A.2   Linear and Exponential Utilities satisfy the properties of Lemma 2

**Independence Property**   Any utility $U_f$ verifies the independence property. Let $\nu_1, \nu_2, \nu_3 \in \mathscr{P}(\mathbb{R})$, $\lambda \in [0, 1]$. Assume $U_f(\nu_1) \geq U_f(\nu_2)$. Then,

$$U_f(\lambda \nu_1 + (1 - \lambda)\nu_3) = \int f \mathrm{d}(\lambda \nu_1 + (1 - \lambda)\nu_3)$$

$$= \lambda \underbrace{\int f \mathrm{d}\nu_1}_{U_f(\nu_1)} + (1 - \lambda) \int f \mathrm{d}\nu_3$$

$$\geq \lambda \underbrace{\int f \mathrm{d}\nu_2}_{U_f(\nu_2)} + (1 - \lambda) \int f \mathrm{d}\nu_3$$

$$= \int f \mathrm{d}(\lambda \nu_2 + (1 - \lambda)\nu_3)$$

$$= U_f(\lambda \nu_2 + (1 - \lambda)\nu_3)$$

In particular, the mean and the exponential utility do.

**Translation Property**   This property comes from the linearity of the mean and the multiplicative morphism of the exponential. Let $\nu_1, \nu_2 \in \mathscr{P}(\mathbb{R})$, $c \in \mathbb{R}$. Assume that $U_{\exp}(\nu_1) \geq U_{\exp}(\nu_2)$ and $U_{\mathrm{mean}}(\nu_1) \geq U_{\mathrm{mean}}(\nu_2)$. Then,

$$U_{\exp}(\tau_c \nu_1) = \int \exp(r) \mathrm{d}\tau_c \nu_1(r)$$

$$= \int \exp(r + c) \mathrm{d}\nu_1(r)$$

$$= \exp(c) \int \exp(r) \mathrm{d}\nu_1(r)$$

$$= \exp(c) U_{\exp}(\nu_1)$$

$$\geq \exp(c) U_{\exp}(\nu_2)$$

$$= \ldots$$

$$= U_{\exp}(\tau_c \nu_2)$$

$$U_{\text{mean}}(\tau_c\nu_1) = \int \lambda\tau + b + c\mathrm{d}\tau$$
$$= c + U_{\text{mean}}(\nu_1)$$
$$\geq c + U_{\text{mean}}(\nu_2)$$
$$= \dots$$
$$= U_{\text{mean}}(\tau_c\nu_2)$$

### A.3 Policy Evaluation for linear combinations of moments

Rowland et al. [2019] prove a necessary condition on Bellman-closed utilities, namely that they should be a family of the form $\{x \mapsto x^\ell \exp(\lambda x)|0 \leq \ell \leq L\}$ for the undiscounted, finite-horizon setting. In the discounted case, the necessary condition is only valid for $\lambda = 0$, that is, without the exponential. They also prove that moments (without the exponential) also verify the sufficient condition such that in that setting they are the only Bellman-closed families of utilities.

In the undiscounted setting, to the best of our knowledge, a similar result has not yet been proved. We provide here the sufficient condition for families of the form $\{x \mapsto x^\ell \exp(\lambda x)|0 \leq \ell \leq L\}$. We show that they are Bellman-closed and that this implies that they can be computed by DP.

Let's consider the family $s_k(\nu) = \int r^k \exp(\lambda r)\mathrm{d}\nu(r)$ for $k \in [n]$ and some fixed $\lambda \in \mathbb{R}$.

$s_n(\eta_h^\pi(x, a))$

$= \mathbb{E}\left[Z_h^\pi(x, a)^n \exp(\lambda Z_h^\pi(x, a))\right]$

$= \mathbb{E}\left[(R_h(x, a) + Z_{h+1}^\pi(X', A'))^n \exp(\lambda(R_h(x, a) + Z_{h+1}^\pi(X', A')))\right]$

$= \mathbb{E}_{x',a'}\left[\mathbb{E}_{R_h, Z_{h+1}}\left[(R_h(x, a) + Z_{h+1}^\pi(x', a'))^n \exp(\lambda(R_h(x, a) + Z_{h+1}^\pi(x', a')))|X' = x', A' = a'\right]\right]$

$= \sum_{x',a'} p_h(x, a, x')\pi_h^{x'}(a')\mathbb{E}_{R_h, Z_{h+1}}\left[(R_h(x, a) + Z_{h+1}^\pi(x', a'))^n \exp(\lambda(R_h(x, a) + Z_{h+1}^\pi(x', a')))\right]$

$= \sum_{x',a'} p_h(x, a, x')\pi_h^{x'}(a')\mathbb{E}_{R_h, Z_{h+1}}\left[\sum_{k=0}^n \binom{n}{k} R_h(x, a)^{n-k} \exp(\lambda R_h(x, a)) Z_{h+1}^\pi(x', a')^k \exp(\lambda Z_{h+1}^\pi(x', a'))\right]$

$= \sum_{x',a'} p_h(x, a, x')\pi_h^{x'}(a') \sum_{k=0}^n \binom{n}{k}\mathbb{E}_{R_h}\left[R_h(x, a)^{n-k} \exp(\lambda R_h(x, a))\right]\mathbb{E}_{Z_{h+1}}\left[Z_{h+1}^\pi(x', a')^k \exp(\lambda Z_{h+1}^\pi(x', a'))\right]$

$= \sum_{x',a'} p_h(x, a, x')\pi_h^{x'}(a') \sum_{k=0}^n \binom{n}{k}\mathbb{E}_{R_h}\left[R_h(x, a)^{n-k} \exp(\lambda R_h(x, a))\right] s_k(\eta_{h+1}^\pi(x', a'))$

This first proves that this family of statistical functional is Bellman closed: they can be expressed as a linear combination of the others. Moreover, on the right-hand side, the expression only depends on the distributions and functionals at the current step $h$ and at the next step $h + 1$. Thus, it provides a natural way to evaluate these functionals by DP.

## B Categorical Projection: an alternative parametrization

The categorical projection was proposed and studied in Bellemare et al. [2017], Rowland et al. [2018], Bellemare et al. [2023]. For a bounded return distribution, it spreads a fixed number $N$ of Diracs evenly over the support and used weight parameters to represent the true distribution. The parameter $N$ is often referred to as the *resolution* of the projection. More precisely, on a support $[V_{\min}, V_{\max}]$, we write $\Delta = \frac{V_{\max} - V_{\min}}{N-1}$ the step between atoms $z_i = V_{\min} + i\Delta, \quad i \in [\![0, N-1]\!]$. We define the projection of a given Dirac distribution $\delta_y$ on the parametric space $\mathscr{P}_C(\mathbb{R}) = \{\sum_i p_i\delta_{z_i}|\, 0 \leq p_i \leq 1, \sum_i p_i = 1\}$ by

$$\Pi_C(\delta_y) = \begin{cases} \delta_{z_0} & y \leq z_0 \\ \frac{z_{i+1}-y}{z_{i+1}-z_i}\delta_{z_i} + \frac{y-z_i}{z_{i+1}-z_i}\delta_{i+1} & z_i < y < z_{i+1} \\ \delta_{z_{N-1}} & y \geq z_{N-1} \end{cases} \tag{11}$$

This definition can naturally be extended to any bounded distribution $\nu$, and by extension, $\Pi_C \eta = (\Pi_C \eta(s,a))_{(s,a) \in \mathcal{X} \times \mathcal{A}}$. This linear operator minimizes the Cramér distance of the parametrization to the parametric space [Rowland et al., 2018].

This projection verifies this approximation bound, analog to the quantile projection,

$$\sup_{(x,a) \in \mathcal{X} \times \mathcal{A}} \ell_2(\Pi_C \eta(x,a), \eta(x,a)) \leq \frac{\Delta_\eta}{N} . \tag{12}$$

Using the property that $W_1(\nu_1, \nu_2) \leq \sqrt{\Delta_\eta} \ell_2(\nu_1, \nu_2)$, the results with the quantile projection can be adapted for the categorical projection, adding $\sqrt{\Delta_\eta}$ factors to the bounds.

This parametrization has the nice property of preserving the mean of the distribution. Yet, even for risk-neutral RL, Quantile-based DistRL algorithm seem to work better and display better properties[Dabney et al., 2018b,a].

## C  Proofs for Policy Evaluation with parameterized distributions

### C.1  Proof of Proposition 1

We recall the statement of Proposition 1: Let $\pi$ be a policy and $\eta^\pi$ the associated Q-value distributions. Assume the return is bounded on a interval of length $\Delta_\eta \leq H\Delta_R$, where $\Delta_R$ is the support size of the reward distribution. Let $\hat{\eta}^\pi$ be the Q-value distributions obtained by dynamic programming (Algorithm 1) using the quantile projection $\Pi_Q$ with resolution $N$. Then,

$$\sup_{(x,a,h) \in (\mathcal{X}, \mathcal{A}, [H])} W_1(\hat{\eta}_h^\pi(x,a), \eta_h^\pi(x,a)) \leq H\frac{\Delta_\eta}{2N} \leq H^2 \frac{\Delta_R}{2N} .$$

To avoid clutter of notation, we denote $\overline{W}_1(\hat{\eta}, \eta) := \sup_{(x,a) \in \mathcal{X} \times \mathcal{A}} W_1(\hat{\eta}(x,a), \eta(x,a))$.

*Proof.* First recall that for any Q-value distribution $(\eta_h)_{h \in [H]}$, with the return bounded on an interval of length $\Delta_\eta \leq H\Delta_R$, and $\Pi$ one of the projection operator of interest with resolution $n$, we have the following bound on the projection estimation error due to Rowland et al. [2019] (Eq (6)):

$$\overline{W}_1(\Pi\eta, \eta) \leq \frac{\Delta_\eta}{2N} . \tag{13}$$

At a fixed step $h \in [H]$, we have the following inequality:

$$\begin{aligned}
\overline{W}_1(\hat{\eta}_h^\pi, \eta_h^\pi) &= \overline{W}_1(\Pi \mathcal{T}_h^\pi \hat{\eta}_{h+1}^\pi, \mathcal{T}_h^\pi \eta_{h+1}^\pi) \\
&\leq \overline{W}_1(\Pi \mathcal{T}_h^\pi \hat{\eta}_{h+1}^\pi, \mathcal{T}_h^\pi \hat{\eta}_{h+1}^\pi) + \overline{W}_1(\mathcal{T}_h^\pi \hat{\eta}_{h+1}^\pi, \mathcal{T}_h^\pi \eta_{h+1}^\pi) \tag{14} \\
&\leq H\frac{\Delta_R}{2N} + \overline{W}_1(\hat{\eta}_{h+1}^\pi, \eta_{h+1}^\pi) . \tag{15}
\end{aligned}$$

Where (14) is due to the triangular inequality with $\mathcal{T}_h^\pi \hat{\eta}_{h+1}^\pi$ as the middle term. In (15), the first term comes from applying (13) to the first term of the previous line. The second term is a consequence of the non-expansive property of the Bellman operator [Bellemare et al., 2017]:

$$\overline{W}_1(\mathcal{T}\eta_1, \mathcal{T}\eta_2) \leq \overline{W}_1(\eta_1, \eta_2) .$$

Using it recursively starting from $h = 1$, and using the fact that $\hat{\eta}_H^\pi = \eta_H^\pi$ we get:

$$\overline{W}_1(\hat{\eta}_1^\pi, \eta_1^\pi) \leq H\frac{\Delta_R}{2N} + \overline{W}_1(\hat{\eta}_2^\pi, \eta_2^\pi) \leq 2H\frac{\Delta_R}{2N} + \overline{W}_1(\hat{\eta}_3^\pi, \eta_3^\pi) \leq \cdots \leq H^2 \frac{\Delta_R}{2N} .$$

$\square$

### C.2 Proof of Lemma 1

We recall the statement of Lemma 1: Let $s$ be either a utility or a distorted mean and let $L$ be the Lipschitz coefficient of its characteristic function. Let $\nu_1, \nu_2$ be return distributions. Then:

$$|s(\nu_1) - s(\nu_2)| \leq L W_1(\nu_1, \nu_2) \,.$$

*Proof.* We prove the property for each family of utilities separately:

**Case 1:** $s$ is a utility. There exists $f$ such that $s(\nu) = \int f \mathrm{d}\nu$. Let $L_f$ be its Lipsichtz constant. The Kantorovitch-Rubenstein duality [Villani, 2003] states that:

$$W_1(\nu_1, \nu_2) = \frac{1}{L_f} \sup_{||g||_L \leq L_f} \left( \int g \, \mathrm{d}\nu_1 - \int g \, \mathrm{d}\nu_2 \right) , \tag{16}$$

where $|| \cdot ||_L$ is the Lipschitz norm. We then immediatly get:

$$L_f W_1(\nu_1, \nu_2) \geq \left| \int f \, \mathrm{d}\nu_1 - \int f \, \mathrm{d}\nu_2 \right| = |s(\nu_1) - s(\nu_2)| \,. \tag{17}$$

**Case 2:** $s$ is a distorted mean. There exists $g$ such that $s(\nu) = \int_0^1 g'(\tau) F_\nu^{-1}(\tau) \mathrm{d}\tau$. Let $L_g$ be its Lipschitz coeffecient. Thus:

$$
\begin{aligned}
|s(\nu_1) - s(\nu_2)| &= \left| \int_0^1 g'(\tau) \left( F_{\nu_1}^{-1} - F_{\nu_2}^{-1}(\tau) \right) \mathrm{d}\tau \right| \\
&\leq ||g'||_\infty \int_0^1 \left| F_{\nu_1}^{-1}(\tau) - F_{\nu_2}^{-1}(\tau) \right| \mathrm{d}\tau \\
&\leq L_g W_1(\nu_1, \nu_2) \,.
\end{aligned}
$$

$\square$

## D About the tightness of Theorem 1

The upper bound provided by Theorem 1 is mainly based on Proposition 1. The latter is obtained by summing, for every step, the Projection Bound by Rowland et al. [2019] (Eq. 6). Thus, achieving the bound would requires first to find a problem instance for which, at every step, the projection bound is tight. Then it would require to verify that the total error is the sum of those projection errors.

The experiment in Figure 2 already shows that summing the total error is very close to the sum of the projection errors. However, in that example, the projection error bound is not reached after the first step. In the following, we exhibit an MDP for which the projection bound is tight at every timestep.

First, let us consider a family of distributions for which the projection error is tight:

**Proposition 2.** *Let $N \in \mathbb{N}$, $\Delta \in \mathbb{R}^+$. Consider $z_i = \frac{i\Delta}{N}$. The distribution*

$$\nu_{N,\Delta} = \frac{1}{2N} \sum_{i=0}^{N-1} (\delta_{z_i} + \delta_{z_{i+1}})$$

*has a support of length $\Delta$ and verifies $W_1(\nu_{N,\Delta}, \Pi_Q \nu_{N,\Delta}) = \frac{\Delta}{2N}$ .*

*Proof.* The cumulative distribution function (CDF) the distribution $\nu_{N,\Delta}$ is

$$
F_{\nu_{N,\Delta}}(x) = \left\{
\begin{array}{ll}
0 & x < 0 = z_0 \,, \\
\frac{2i+1}{2N} = \tau_i & z_i \leq x < z_{i+1} \,, \\
1 & x \geq 1 = z_N \,.
\end{array}
\right.
$$

We write $\tau_i = \frac{2i+1}{2N}$, so that $\forall i \in [|0, N-1|], F_{\nu_{N,\Delta}}(z_i) = \tau_i$. We now explicit the projection of $\nu_{N,\Delta}$ and the Wasserstein distance relative to it. A possible Quantile projection is $\Pi_Q \nu_{N,\Delta} = \frac{1}{N} \sum_{i=0}^{N-1} \delta_{z_i}$,

and

$$W_1(\nu_{N,\Delta}, \Pi_Q \nu_{N,\Delta}) = \int_0^1 |F_\nu^{-1}(w) - F_{\Pi_Q \nu}^{-1}(w)| dw$$

$$= \sum_{i=0}^{N-1} \int_{\frac{i}{N}}^{\frac{i+1}{N}} |F_\nu^{-1}(w) - \underbrace{F_{\Pi_Q \nu}^{-1}(w)}_{z_i}| dw$$

$$= \sum_{i=0}^{N-1} \int_{\frac{i}{N}}^{\tau_i} |\underbrace{F_\nu^{-1}(w)}_{z_i} - z_i| dw + \int_{\tau_i}^{\frac{i+1}{N}} |\underbrace{F_\nu^{-1}(w)}_{z_{i+1}} - z_i| dw$$

$$= \sum_{i=0}^{N-1} \frac{1}{2N} \underbrace{(z_{i+1} - z_i)}_{\frac{\Delta}{N}}$$

$$= \sum_{i=0}^{N-1} \frac{\Delta}{2N^2}$$

$$= \frac{\Delta}{2N}$$

$\square$

The value distribution of the following time steps is obtained by applying two operators: the Bellman operator and the projection operator. Here will consider a MDP with only one state. We need to find such operators so that $\Pi_Q \mathcal{T} \nu_{N_1 \Delta_1} = \nu_{N_2 \Delta_2}$, where the Bellman operator simply consists in the added reward distribution.

**Proposition 3.** *Let $N \in \mathbb{N}$, $\Delta \in \mathbb{R}^+$. Consider $\varrho = \frac{1}{2}(\delta_0 + \delta_{\frac{\Delta}{N-1}})$. Then there exists a Quantile projection operator $\Pi_Q$ such that*

$$\varrho * \Pi_Q(\nu_{N,\Delta}) = \nu_{N,(\Delta + \frac{\Delta}{N-1})}$$

*Proof.* We consider $\Pi_Q(\nu_{N,\Delta}) = \frac{1}{N} \sum_{i=0}^{N-1} \delta_{\frac{i\Delta}{N-1}}$. Since $\forall i, \frac{i\Delta}{N-1} \leq z_i < \frac{(i+1)\Delta}{N-1}$, we have $F(z_i) = \tau_i$, verifying that it is indeed a valid Quantile Projection.

Hence,

$$\varrho * \Pi_Q(\nu_{N,\Delta}) = \frac{1}{2}(\delta_0 + \delta_{\frac{\Delta}{N-1}}) * \left( \frac{1}{N} \sum_{i=0}^{N-1} \delta_{\frac{i\Delta}{N-1}} \right)$$

$$= \frac{1}{N} \sum_{i=0}^{N-1} \left( \frac{1}{2}(\delta_0 + \delta_{\frac{\Delta}{N-1}}) * \delta_{\frac{i\Delta}{N-1}} \right)$$

$$= \frac{1}{2N} \sum_{i=0}^{N-1} \left( \delta_{\frac{i\Delta}{N-1}} + \delta_{\frac{(i+1)\Delta}{N-1}} \right)$$

$$= \nu_{N,(\Delta + \frac{\Delta}{N-1})}$$

The last egality comes down to noticing that $\frac{i\Delta}{N-1} = \frac{i}{N}(\Delta + \frac{\Delta}{N-1})$. $\square$

Here, we take advantage of the fact that the Quantile Projection is not unique (see discussion in section 2.1). By choosing the adapted projection, it is then possible to obtain one of those distribution at every timestep.

**Corollary 1.** *Let $N \in \mathbb{N}$, $\Delta_0 \in \mathbb{R}^+$. Consider the sequence $\Delta_{h+1} = \Delta_h(1 + \frac{1}{N-1})$ and $\varrho_h = \frac{1}{2}(\delta_0 + \delta_{\frac{\Delta_h}{N-1}})$. Consider $\Pi_Q$ as in Prop. 3.*

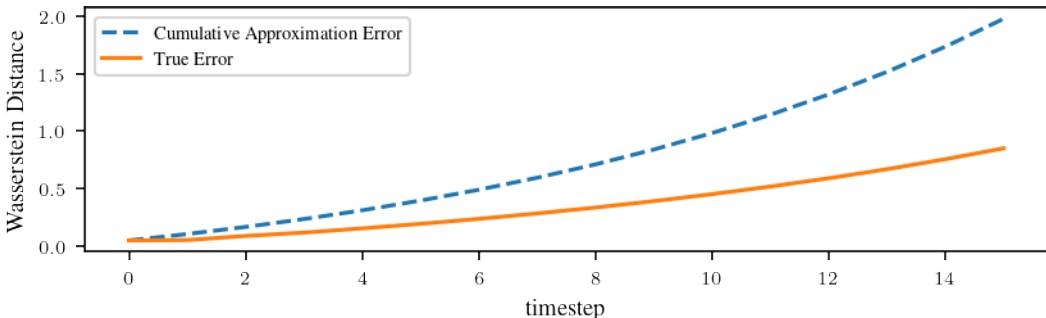

Figure 3: Evaluation of the Wasserstein Distance between the true value distribution and the approximated one, in the MDP described in Corollary 1

.

*Consider the MDP with only one state $x$ and action $a$, reward distribution $\varrho_h$, horizon $H$. Consider $\hat{\eta}_h$ the value distributions obtained throught dynamic programming with quantile projection. Then, at any timestep $h$, the error induced by the projection operator matches the bound in Eq. 6:*

$$W_1(\Pi_Q \eta_h, \eta_h) = \frac{\Delta_h}{2N}$$

While the projection error is maximal in such instance, it was verified experimentally that the bound in Prop. 1 was still not tight. This comes from the fact that the Bellman operator is not a non-contraction is this case, and that the triangular inequality used to sum the projection error is not tight either.

We hence found that every inequality used in proving Theorem 1 can be tight, but there does not seem to exist an instance for which all the inequalities are tight at the same time, meaning that this bound would be never reached exactly.

# E   Proof of the main results

The proof of the result is divided in parts. First we show that Bellman optimizable functionals verify the two properties of Lemma 2 (Independence and Translation). Then, using those properties, we prove that Bellman optimizable functionals can only be exponential utilities (Theorem 2). Using the known fact that exponential utilities are bellman optimizable, we obtain the full characterization.

## E.1   Proof of Lemma 2

*Proof.* **To prove that each property is necessary**, we use a proof by contradiction, and exhibit MDPs where the algorithm is not optimal when the property is not verified.

**Independence Property**    Let $s$ be a Bellman optimizable statistical functional that does not satisfy the Independence property. That is, there exists $\nu_1, \nu_2, \nu_3 \in \mathscr{P}(\mathbb{R})$ and $\lambda \in [0,1]$ such that $s(\nu_1) \geq s(\nu_2)$ but $s(\lambda\nu_1 + (1-\lambda)\nu_3) < s(\lambda\nu_2 + (1-\lambda)\nu_3)$ .

Then consider the MDP in Fig.4 (left) with horizon $H = 2$ corresponding to the depth of the tree: The agent starts in Start and must take 2 actions, a unique but random and non-rewarding one ($a_0$) and a final deterministic step ($a_1$ or $a_2$) to a rewarding state. Thus, by construction, the optimal strategy is $(a_0, a_2)$ that leads to End 2 with probability $\lambda$ (and End 3 with probability $1 - \lambda$). The true optimal distribution at Start state is $\eta_0^* = \lambda\nu_2 + (1-\lambda)\nu_3$. We compute the distributions output by the algorithm:

$$
\begin{aligned}
H = 2: &\quad \eta_2(\text{End 1}, a^*) = \delta_0, \quad \eta_2(\text{End 2}, a^*) = \delta_0, \quad \eta_2(\text{End 3}, a^*) = \delta_0 \\
H = 1: &\quad \eta_1(\text{Left}, a^* = \arg\max_a s(\nu_a)) = \nu_1, \quad \eta_2(\text{Right}, a^* = a_1, a_2) = \nu_3 \\
H = 0: &\quad \eta_0(\text{Start}, a^* = a_0) = \lambda\nu_1 + (1-\lambda)\nu_3
\end{aligned}
$$

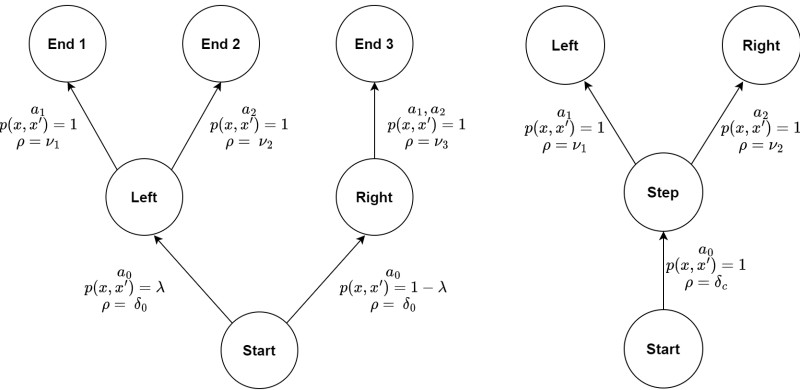

Figure 4: Left: Independence Property Counter Example, Right: Translation Property Counter Example. Each arrow represents a state transition, which is characterized by the action leading to the transition, the probability of such transition, and the reward distribution of the transition.

The output return distribution $\eta_0$ is not the true optimal $\eta_0^*$ for $s$ so the algorithm is incorrect which is a contradiction as $s$ is assumed to be Bellman optimizable. Hence the property is needed.

**Translation Property** Let $s$ be a Bellman optimizable statistical functional that does not verify the Translation Property, *i.e.* there exists $\nu_1, \nu_2 \in \mathscr{P}(\mathbb{R})$, $c \in \mathbb{R}$ such that $s(\nu_1) \geq s(\nu_2)$ but $s(\tau_c \nu_1) < s(\tau_c \nu_2)$. Then consider MDP in Fig.4 (right). The optimal strategy is again $(a_0, a_2)$ by construction. The algorithm output the following distribution:

$$
\begin{aligned}
H = 2: \quad & \eta_2(\text{Left}, a^*) = \delta_0, \quad \eta_2(\text{Right}, a^*) = \delta_0 \\
H = 1: \quad & \eta_1(\text{Step}, a^*) = \nu_1 \\
H = 0: \quad & \eta_0(\text{Start}, a^*) = \tau_c \nu_1
\end{aligned}
$$

So here again, the algorithm does not output an optimal distribution for $s$, hence the necessity of the property.

This proof shows that both properties are necessary, but not that they are sufficient. The other implication could be proven, but the proof would be unnecessary as those properties are enough to restrict to only 1 class of function for which we already know is bellman optimizable.

$\square$

### E.2 Proof of Theorem 2

*Proof.* For any return distribution $\nu$, let $s(\nu)$ be the considered functional. Since we assume that $s$ is $W_1$-continuous Bellmman optimizable, the by the Independence Property (Lemma 2) we know from the Expected Utility Theorem that we can assume: $s(\nu) = s_f(\nu) = \int_{\mathbb{R}} f(x) d\nu(x)$ for some continuous, monotonous mapping $f$. Without loss of generality, we may assume by a density argument that $f$ is twice continuously differentiable.

By the Intermediate Value Theorem, we can then define $\phi(h) = f^{-1}(\frac{1}{2}(f(0) + f(h)))$, so that $\frac{1}{2}(f(0) + f(h)) = f(\phi(h))$ and in particular $\phi(0) = f^{-1}(f(0)) = 0$.

A very special case is when $f$ is constant, which satisfies the theorem. We now assume that there exists point $x_0$ such that $f'$ does not vanish on a neighborhood of $x_0$. On this neighborhood, using the inverse function theorem, $\phi$ is also twice differentiable. Without loss of generality, we assume that $x_0 = 0$.

For any fixed $h > 0$, we consider the probability distributions $\nu_1 = \frac{1}{2}(\delta_0 + \delta_h)$ and $\nu_2 = \delta_{\phi(h)}$. Remark that $s_f(\nu_1) = \int f \, d\nu_1 = \frac{1}{2}(f(0) + f(h))$ and $s_f(\nu_2) = f(\phi(h)) = \frac{1}{2}(f(0) + f(h))$ by definition of $\phi$, so $s_f(\nu_1) = s_f(\nu_2)$.

The Translation property implies that for all $x \in \mathbb{R}$, $s_f(\nu_1) \leq s_f(\nu_2) \implies s_f(\nu_1(\cdot + x)) \leq s_f(\nu_2(\cdot + x))$ and $s_f(\nu_1) \geq s_f(\nu_2) \implies s_f(\nu_1(\cdot + x)) \geq s_f(\nu_2(\cdot + x))$. Hence, $\forall x \in \mathbb{R}$, $\frac{1}{2}(f(x) + f(x+h)) = f(x + \phi(h))$.

This equation can be differentiated twice with respect to $h$. For any value of $x$, we obtain:

$$\frac{1}{2}f'(x + h) = \phi'(h)f'(x + \phi(h)) \quad \text{and} \tag{18}$$

$$\frac{1}{2}f''(x + h) = \phi''(h)f'(x + \phi(h)) + \phi'(h)^2 f''(x + \phi(h)) . \tag{19}$$

Recall that by definition, $\phi(0) = 0$. Now, for $x = 0$, Eq. (18) yields $\frac{1}{2}f'(0) = \phi'(0)f'(\phi(0)) = \phi'(0)f'(0)$ and, since $f'(0) \neq 0$, $\phi'(0) = \frac{1}{2}$.

Now, choosing $h = 0$ in (19) and plugging in the values of $\phi(0)$ and $\phi'(0)$, we obtain for all $x \in \mathbb{R}$:

$$\frac{1}{4}f''(x) = \phi''(0)f'(x) .$$

We then consider two cases, depending on whether $\phi''(0)$ is null or not.

**Case 1:** $\phi''(0) = 0$. The equation simply becomes $f''(x) = 0$, hence $f$ is affine: $\exists\, a, b \in \mathbb{R}, \ f(x) = ax + b$.

**Case 2:** $\phi''(0) \neq 0$. We write $\beta = 4\phi''(0)$. The differential equation becomes $f''(x) = \beta f'(x)$, whose solutions are of the form

$$\exists\, c_1, c_2, \beta, \quad f(x) = c_1 \exp(\beta x) + c_2 .$$

Hence, $f$ can only be the identity or the exponential, up to an affine transformation.

$\square$

