# OpenReview forum: "Beyond Average Return in Markov Decision Processes"
_NeurIPS.cc/2023/Conference — NeurIPS 2023 poster_

### Official Review · Reviewer_3T3p · 2023-07-07

**Soundness:** 3 good
**Presentation:** 3 good
**Contribution:** 2 fair
**Rating:** 6
**Confidence:** 3

**Summary:**

The paper investigates general functionals of the distribution over returns in three situations:
1) For policy evaluation, the authors prove some error bounds for a given distributional RL algorithm.
2) For planning, the authors exhibit the functional form that is optimizable via dynamic programming.
3) For reinforcement learning, they provide a Q-learning algorithm when the functional is linear or exponential

**Strengths:**

The paper investigates an interesting question, which is useful for sequential decision-making with more sophisticated decision criteria.

The writing is generally quite clear.

**Weaknesses:**

The obtained error bound for policy evaluation seems to be very loose, as also suggested by the experiments. Can we say anything about the tightness of this bound?

The novelty of some of the obtained results seems to be limited:
1) How different is the known result about Bellman closedness and the new result about Bellman optimizability (Theorem 2)? Isn't the former property a necessary condition for the latter?
2) Regarding the proposed Q-learning algorithm, the linear case seems to be trivial. For the exponential case, how different is the proposed update compared to Borkar's?

The paper should be checked for typos (see below for some of them).

**Questions:**

See questions above.

Some minor issues:
- line 52: p_h -> p_h^a(x, \cdot) and R_h -> R_h(x, a)
- line 74: p_{h+1}^a(s, \cdot)  -> p_{h+1}^a(x, \cdot)
- line 95: the definition of \mathcal P_Q(\mathbb R) should include the condition in (5)
- line 96: W_1 should be defined here
- line 209: m
- line 222: \eta vs \eta^*
- line 227: missing /extra word?
- line 239: the other von Neumann-Morgenstern axioms are not needed?
- line 269: sections addresses
- line 271: linear of exponential

---

> ### Author Rebuttal · Authors · 2023-08-09
>
> Thank you for your supportive and insightful feedback. Please see our response addressing your concerns and questions :
>
> - The obtained error bound for policy evaluation seems to be very loose, as also suggested by the experiments. Can we say anything about the tightness of this bound?
>
> First,as explained in the common response,  we found a mistake , which led to inconsistent Wasserstein distance computation and made our bound look much worse than it actually is. This was solved, and the corrected graphics can be found in the global rebuttal.
>
> Concerning the tightness of the bound, this was investigated more in detail after the submission and added to the appendix later. Our first find is that this bound is never reached exactly for $H \geq 1$. Yet, we believe it might not be possible to refine it in general. Indeed, the bound is proven through a main idea that is summing the successive error bounds due to the projection operator. We have made two important observations :
>
> First, it is possible to maximize the projection error bound at each step (meaning, at every step h, the error due to the projection operator matches the bound of Eq.6). This implies that the projection bounds cannot be improved, even after successive steps in general.
>
> Then the experiment of the paper with the corrected computation shows that in a simple environment, the sum of the successive projection errors matches very closely the overall error between the true distribution and the approximated one. This implies that the method of the proof is relevant, and may not be improved as well.
>
> Combining those two findings, we could not find an immediate improvement and we are ready to conjecture it cannot be significantly improved in general.
>
>  - How different is the known result about Bellman closedness and the new result about Bellman optimizability (Theorem 2)? Isn't the former property a necessary condition for the latter?
>
> The idea of Bellman Closedness is to find a Bellman Equation. We want to compute the values of the statistics on some state and action recursively, with the values on the other states and actions. That is, we try to find a general formula to write $s(r + PV)$ as a function of $r$ the immediate reward, $P$ the transitions and $s(V)$ where $V$ is the value function and $s$ the statistic. In short, Bellman Closedness is all about computing the values by themselves, by recursion.
>
> With Bellman optimizability, we check instead if a policy improvement algorithm works and finds the optimal policy, without the concern of computing such values. We assume that we can compute the distribution exactly, and thus have access to the exact values of statistics. We assume that for each state and a current policy, which action maximizes the chosen statistics of the value function. The question we answer is : “Assuming we have access to the value functions for the statistic, will a policy improvement algorithm converge to an optimal policy?”
>
> - Regarding the proposed Q-learning algorithm, the linear case seems to be trivial. For the exponential case, how different is the proposed update compared to Borkar's?
>
> In this section, we do not propose any new algorithm, both already exist and are mentioned here to recall to the viewer that our theoretical find matches the known algorithm, and to complete the full picture addressed by the paper, that is statistics in different MDP problems : Policy Evaluation, Planning, and finally Reinforcement Learning. This is also addressed in the global rebuttal.
>
>    We also thank you for pointing out typos and writing errors, we have now fixed them all and made a thorough grammatical pass on the paper.

---

> > ### Comment · Reviewer_3T3p · 2023-08-16
> > **Response**
> >
> > Thank you for the rebuttal. I am still not clear about the relation between Bellman closedness and Bellman optimizability. Should Bellman close functional necessarily be Bellman optimizable?

---

> > > ### Author Response · Authors · 2023-08-18
> > >
> > > Thank you again for the review,
> > >
> > > No, Bellman Closed functionals should not necessarily be Bellman Optimizable. An example would be the Variance, or even the utility function $x\exp (x)$, which are both in Bellman closed sets but not Bellman Optimizable. This has now been clarified in the paper to emphasize on the difference between the two concepts.

---

### Official Review · Reviewer_kE1T · 2023-07-07

**Soundness:** 3 good
**Presentation:** 3 good
**Contribution:** 3 good
**Rating:** 6
**Confidence:** 3

**Summary:**

This paper considers the problem of using distributional reinforcement learning to perform policy evaluation and planning with more general classes of reward functionals than those typically considered in standard Markov decision processes and RL. The paper focuses in particular on dynamic programming methods for obtaining distributional Q functions based on distributional Bellman operators, as well as how to use these methods for planning with the resulting distributional Q functions. The main contributions of the paper are twofold: (1) when performing distributional policy evaluation on general utilities using a certain kind of distributional Q function approximation scheme, worst-case bounds in terms of Wasserstein distance of the approximation from the true distributional Q function are provided; (2) when performing planning using distributional Q functions, a new notion of "Bellman optimizability" of reward functionals is given, and it is shown that the only Bellman optimizable reward functionals are the class of exponential utilities. It was previously known that standard (i.e., not distributional) dynamic programming methods (see, e.g., lines 213-214) can be applied to solve planning problems with general utilities, so contribution (2) provides a negative result suggesting that distributional RL methods, though providing a very general framework for performing evaluation and planning via dynamic programming, are not strictly necessary to solve the class of problems to which they are theoretically suited.

**Strengths:**

Overall, the paper is well-written and the results add clarity to the distributional RL landscape that is likely to be of interest to the community. Regarding significance, contribution (1) outlined in the summary given above is potentially useful for a wider range of utilities than previously considered, but the results appear to be straightforward extensions of existing results (see C.1, C.2 in the appendix). The main result of the paper is instead contribution (2), provided in the form of Theorem 2 (lines 248-250). This result delimits the applicability of distributional dynamic programming to classes of utilities that can be handled using non-distributional techniques, suggesting that the primary usefulness of distributional RL methods may be experimental instead of theoretical. It is refreshing to see an ML paper making an effort to soberly clarify the mathematical limits of a particular subfield of ML research. The proof of this result (see D.2) appears to boil down to first constructing a differential equation based on the integrand of the utility, then solving to show that the integrand must correspond to an exponential utility. The proof is not long, but the idea is clever.

**Weaknesses:**

I have two main concerns regarding the paper:
* Contributions (1) and (2) are somewhat disjointed. On the one hand, (1) provides reassurances that Algorithm 1 enjoys approximation bounds for a wide variety of different utilities. At the same time, (2) undermines this result by showing that planning using the resulting distributional Q functions is only theoretically meaningful for a much-restricted subset of those utilities. Some clarification of the implications of (2) for (1) would be helpful.
* In the proof of Theorem 2, it is assumed that the utility integrand $f$ is twice-differentiable and $f'(x) \neq 0$, for all values of $x$. This assumption is critical to Theorem 2, thus to contribution (2), and thus to the overall contribution of the paper. It is stated on lines 516-517 that this assumption "could be proven through long and fastidious analysis that is beyond the scope of this article", yet no indication of how this might be accomplished is provided. Given its importance to the paper, I argue that this is within scope of the article, so that a proof sketch or at least a convincing explanation of why this assumption holds is needed.

**Questions:**

* Given Theorem 2, what is the main usefulness and impact of the approximation bounds provided in section 3?
* Why is Bellman optimizability (Definition 2, lines 221-222) the right notion of what it means for a problem to be solvable using distributional RL? Are there incompatible notions of optimality that might invalidate Theorem 2? If not, why not?
* Why does the assumption regarding the utility integrand $f$ on lines 516-517 in the appendix hold?
* It is mentioned that policy stationarity is disrupted in the discounted setting on lines 289-290. Can you provide references ensuring sufficiency of stationary policies in the setting you consider?

**Limitations:**

Aside from the weaknesses and questions above, the authors have adequately addressed the limitations of this work.

---

> ### Author Rebuttal · Authors · 2023-08-09
>
> Thank you for your supportive and insightful feedback. Please see our response addressing your concerns and questions:
>
> > About the relevance of section 3 related to theorem 2:
>
> This matter was addressed in the global rebuttal as it asked by several reviewers
>
> > About the definition of Bellman Optimizabitily:
>
> This is also addressed in the global rebuttal for the same reasons
>
> > Are there incompatible notions of optimality that might invalidate Theorem 2? If not, why not?
>
> The question on the existence of other notions of ‘optimality’ is unclear to us. An optimal policy is always one that has the maximal value (regardless of the goal, mean reward, utility, etc). Perhaps you meant ‘notions of optimizability’?
>
> > About the loss of stationarity in the discounted setting on lines 289-290 :
>
> This is a misunderstanding due to the ambiguity of our phrasing. Another way to write it is the following : ”there exists a method to optimize the exponential utility through dynamic programming in discounted MDPs [Cheung and Sobel, 1987]. This approach requires modifying the functional to optimize at each step. A difference with the risk-neutral objective in discounted MDPs, is that the optimal policy may not be stationary”. We never assumed to have stationary policies in our undiscounted setting.In discounted MDPs, however, it is known that the policy optimizing the expected reward can be chosen to be stationary, and we found relevant to highlight that it is no longer the case when we optimize the exponential utility.
>
>
> > Assumptions in the proof of theorem 2 : Differentiation and non-zero derivative
>
> We agree with you that Theorem 2 uses strong assumptions with no explanations. In fact, the properties that we claim for the utilities can be proved and we have done it for this response. All details have been added to the appendix:
>
> 1. We can reduce the problem to differentiable utilities thanks to an approximation argument.
> 2. Using the continuity and translation property, we can prove that f is either monotone or constant. Similarly, we prove that f’ is monotone.This implies that the derivative is always non-zero.
> We will try to convey the main intuition of the proofs of these results here although the full proof does not fit.
>
> 	- For **the differentiation assumption**, a way to look at it is to consider that any integrable function can be arbitrarily approximated with infinitely-differentiable functions (see [1] p119, about the density of differentiable functions in the L_1 space) . Thus, for any function, there exists a differentiable one that is close enough such that optimizing  the initial function becomes the same as optimizing on this differentiable one. Hence, in the same way the Von Neumann - Morgenstein theorem is used to reduce the study to utilities, this approximation allows us to reduce the study to differentiable utilities.
>
> 	- **The monotonicity and convex/concave property** are proven using the translation property to transform a local inequality in a global one. Start by considering 2 points, for instance $0$ and $1$. We assume that $f(0) < f(1)$ and  show that $f$ is increasing. To do so, we use the translation property with $\nu_1 = \delta_0, \nu_2 = \delta_1$, which gives the following : $\forall c, f(0 + c) \leq f(1 + c)$ (a local inequality becomes a global inequality). In particular by choosing $c$ to be successively $1, 2, n, \dots$, we obtain the chain of inequality $f(0) \leq f(1) \leq f(2) \leq \dots \leq f(n)$. Now we do the same starting with $0$ and $1/2$ and $c$ a multiple of $1/2$, to get the chain $f(0) \leq f(0.5) \leq f(1) \leq f(1.5) \dots$. Halving infinitely by recursion, we obtain that $f$ is monotonous on a dense set of numbers. With density and continuity, $f$ is globally monotonous.
> 	Note that in case of equality on two points, this reasoning proves that $f$ is globally constant. Hence, $f$ is either strictly monotonous or constant.
> 	For the monotonicity of $f'$, we use the same idea and show that for any two points, with $f((x_1 + x_2)/2) \leq (f(x_1) + f(x_2))/2$ (or $\geq$). This is called midpoint convexity and is equivalent to convexity. (Here, for instance, we start with $\nu_1 = \delta_{1/2},\ \nu_2 = (1/2) (\delta_0 + \delta_1)$)
>
> 	- The strict monotonicity implies that the derivative has a constant sign, while the concavity or convexity implies the monotonicity of the derivative. A function both strictly monotonous and of constant sign cannot ever reach 0..
>
> [1] Functional Analysis Sobolev Spaces and Partial Differential Equations, Haim Brezis, 2010

---

> > ### Comment · Reviewer_kE1T · 2023-08-15
> >
> > Thanks for your reply. Regarding Theorem 2, I get the general outline of your argument and am reassured. You mention that you have added the details to the appendix -- have you uploaded it? I am currently unable to find it in the supplementary material.

---

> > > ### Author Response · Authors · 2023-08-17
> > > **We cannot submit a revised version**
> > >
> > > Unfortunately, as per Neurips' CFP: "Authors may not submit revisions of their paper or supplemental material, but may post their responses as a discussion in OpenReview."
> > > Unless we are missing something, we are afraid we can only do our best to convey the main ideas fo the proof on OpenReview. Do you have a question on our proof idea? We apologize if it's not clear enough, but we are happy to try our best to clarify any concern you may have.

---

> > > > ### Comment · Reviewer_kE1T · 2023-08-18
> > > >
> > > > Thanks for the clarification. I think your proof sketch sounds reasonable and do not have further questions.

---

### Official Review · Reviewer_RipE · 2023-07-07

**Soundness:** 3 good
**Presentation:** 3 good
**Contribution:** 4 excellent
**Rating:** 7
**Confidence:** 4

**Summary:**

Classic RL maximizes the (discounted) cumulative reward in Markov Decision Processes (MDPs). This paper studies more general functionals of rewards, such as generalized means. These more general functionals are useful to consider for certain applications, such as those with safety concerns. Furthermore, the study of these more general functionals has been not well understood in the literature.

For finite-horizon prediction problems, this paper 1) argues that only generalized means of rewards can be obtained exactly using dynamic programming and there is no need to resort to learning the entire distribution of return (I am confused about this part, see Questions), and 2) provides an upper bound of estimation error for utility functionals, which are more general than generalized means, when approximated with Quantile Distributional RL. Both 1) and 2) mirror Rowland et al.'s (2019) results, which were developed concerning the discounted setting.

The most important result is for finite-horizon control problems. This paper shows that the only Bellman optimizable functionals (those that can be optimized by applying Dynamic Programming to the distribution of return) are generalized means. Combined with a previous result showing that generalized means can be solved using dynamic programming (no distribution involved), this result shows that the only functionals that are optimizable by Distributional RL can be optimized by Dynamic Programming, which is a much more efficient algorithm.

**Strengths:**

The problem that this paper studies is an important one.

The main result, which shows that the only functionals that are optimizable by Distributional RL can be optimized by Dynamic Programming, is remarkable.

**Weaknesses:**

The writing of the paper needs improvements. Specifically:
1) right now, the presentation of the paper is driven by the technical results that it wants to show, rather than the messages that it wants the readers to know. The paper presents an important result about Distributional RL: the only functionals that are optimizable by Distributional RL can be optimized by Dynamic Programming. As the authors highlighted in line 260, this result is the most important one and it narrows down the avenues to explain its (distributional RL) empirical success (line 310). However, this result is not outspoken throughout the paper, until an inconspicuous discussion on page 8. Why not say your most important result in the Abstract and the Introduction sections?

2) The notations given background section need to be clearly defined:
the notation (z_i)_i needs to be defined.
\delta is not defined (line 96).
\mu is not defined (line 97).
F_\mu is not defined (line 98).
what is \eta in line 100?
\Delta_\eta is not defined in line 101.
Line 124: U_f is not defined.

3) typos: extra m at the end of line 209
line 184: should be "see Figure1 (left)"
line 199: should be "Figure 2 (left)"
the order of the two subfigures in Figure 2 would better be reversed in accordance to the texts.

**Questions:**

"Much of the existing theory is based of discounted MDPs, but many recent efficient RL algorithms with strong guarantees are for finite-horizon, undiscounted setting" What kind of theory do you mean for discounted MDPs? Could you give examples that illustrate this argument?

Could you point to the result by Rowland et al. (2019) that gives equation 6?

Note that other parameterizations exist but are less practical. Could you explain why is this true or refer to other works that show that this is true?

Line 122-123: when lambda > 0, the utility is higher for Gaussian distribution with a larger sigma (higher variance), so isn't this case risk-seeking?

Line 124: you don't need to mention this axiom at this point. I have been wondering why I need to know this until I found it being used in section 4.

Line 152 is confusing to me: do you mean that for any L >= 1, statistics corresponding to the first L+1 moments form a Bellman closed set?

Line 152: Theorem 4.3 by Rowland et al. (2019) only mentioned moment functionals but not exponential functionals. Furthermore, although you prove in A.3 that sets of multiplication of moments functionals and exponential functionals also form a Bellman closed set in the finite horizon setting, you didn't show that these are the only families of functionals that form Bellman closed sets in this setting.

The footnote on page 5: how can the support of return be -h + 1 given that all rewards are non-negative?

Line 215: is the question open for utilities other than exponential ones for control problems?

Line 221: why define Bellman optimizability instead of extending the definition of Bellman closeness to control problems? In addition, it is weird to define a quantity, which you want to optimize, using an algorithm that optimizes it. Do you see other ways that make this definition without resorting to a particular algorithm?

Line 241: why does this result narrow down the family Bellman optimizability functionals to utilities? I mean, literally.

Section 5: why would we want to see a new algorithm without any theoretical/empirical study of the algorithm?

---

> ### Author Rebuttal · Authors · 2023-08-09
>
> Thank you for your in depth review of the paper and the relevant questions that will help clarify it. We now address each raised points :
>
> - About the undiscounted setting theory :
>
> Thank you for pointing out this sentence, we agree it was unclear. The goal of this sentence and the following was to justify our choice of studying the finite-horizon setting, which has an impact on our results (discussed in the last section). We rephrase it as follows:
> “Historically, the theory of RL has been established for discounted MDPs (e.g. [Sutton & Barto, 2020; Watkins & Dayan, 1992; Szepesvári, 2010; Bellemare et al., 2023]) but recently more attention was drawn to the undiscounted, finite-horizon setting [Auer, 2002, Osband et al., 2013, Jin et al.,2018, Ghavamzadeh et al., 2020], for which fundamental questions on the theory of MDPs remain open. In this paper...”
> We also tried to clarify the impact of this choice a bit earlier in the paper and refer to discussions for completeness.
>
> - Precise pointers to Rowland’s work (Eq 6 and Th 4.3 (line 152)) :
>
> Those are actually intermediate results that can be found in the appendix of their paper ([2]). Equation 6 corresponds to Lemma B.2. We also want to point out that [1] contains a similar result with an additional ½ factor in the bound (exercise 5.20). For the theorem 4.3, the result is part of the proof which is divided in two parts. First, they start by reducing the study to the set of functions we discuss (equation 14 in their appendix), and then they show that, because of the discount, the exponential part has to be discarded. In our case, without discount, only the first part still holds and that is what we use. We just need to verify that the set is indeed bellman closed in this setting, which we do in appendix A.3.
>
> - “Line 152 is confusing to me: do you mean that for any L >= 1, statistics corresponding to the first L+1 moments form a Bellman closed set?”
>
> Yes, this set of statistics is a Bellman closed set. This was already true and known in the discounted setting considered by Rowland [2]. The difference in ours is the addition of the exponential term in it.
>
> - Relevance and discussion of other parametrizations:
>
> The only other studied parametrization with theoretical grounds is the Categorical Projection, described and discussed in Appendix B. We omitted it in the main text because it is not essential to our argument. Among its issues is the need for a fixed and known support of the distributions of study and the use of a different metric for the bounds[3]. Moreover, algorithms using the quantile approach display better results [4][5].
>
> - "Is the question open for utilities other than exponential ones for control problems?"
>
> To be precise, whether or not any statistical functional other than the exponential utility can be optimized efficiently in general is still an open question, utility or not. We focused on dynamic programming algorithms and thus on the existence of a Bellman equation to support it. Our claim is that for utilities other than exponential, a policy-improvement type of strategy cannot work. We do not claim that there exists no other algorithm that can compute it, this question remains open and would possibly require a complexity theory analysis, such as proving that it is NP-Hard. This is out of the scope of the paper but is an interesting question. We will clarify that again. In RL, however, the predominance of dynamic programming is fundamental and we believe this justifies that our Theorem remains significant and impactful.
>
> - "Line 241: why does this result narrow down the family of Bellman optimizability functionals to utilities?"
>
> The result does not prevent the existence of non-utilities verifying the property. However, the result implies that studying utilities is enough because if another statistical functional can be optimized by such an algorithm, then there exists a utility that yields the exact same results. Hence, if we know the form of all the utilities that verifies such property, we know the form of all statistical functionals in general.
>
> - "Line 221: why define Bellman optimizability instead of extending the definition of Bellman closedness to control problems?"
>
> This question was raised by several reviewers, so we address this in detail in the global rebuttal.
>
> - "Section 5: why would we want to see a new algorithm without any theoretical/empirical study of the algorithm?"
>
> This is also addressed in the global rebuttal: the algorithm is not new nor ours and we mention it in the paper for completeness of our argument that utilities can be learned.
>
> We also thank you for pointing out typos and writing errors, we have now fixed them all and made a thorough grammatical pass on the paper.
>
> [1] Distributional Reinforcement Learning, Bellemare et al., 2023
> [2] Statistics and Samples in Distributional Reinforcement Learning, Rowland et al. 2019
> [3] An Analysis of Categorical Distributional Reinforcement Learning, Rowland et al. 2018
> [4] Distributional Reinforcement Learning With Quantile Regression, Dabney et al. 2018
> [5] Implicit Quantile Networks for Distributional Reinforcement Learning, Dabney et al. 2018

---

### Official Review · Reviewer_3Rfs · 2023-07-25

**Soundness:** 3 good
**Presentation:** 4 excellent
**Contribution:** 3 good
**Rating:** 7
**Confidence:** 3

**Summary:**

This paper addresses

**Strengths:**

1. The method is very well-presented, with notations, terms and algorithms put in a very clear and understandable fashion. They are introduced without fancy names and terms, which is great.
2. The questions are well-stated and addressed respectively from a theoretical perspective, and they are fundamental problems.
3. The significance of contribution is good. It provides a good starting point to address the statistical functional evaluation problem.

**Weaknesses:**

1. Some statements in the propositions lack references to earlier works, such as quantile functions an CVaR (metric).
2. The learning environment description is lacked.
3. The limitations, potential societal impacts and ethical concerns, and future work are lacking.

**Questions:**

My suggestions has been listed in weaknesses, which I will paraphrase here:
1. Please fix grammatical errors and add proper references on early works, especially for the sections with propositions.
2. Please connect your work to more practical applications. I know it might be hard to conduct experiments as additions to this paper, but some discussions would be helpful.
3. Please give more details about the experiments on functional distance calculation.
4. Most importantly, please add descriptions on potential limitations, concerns, and societal impacts about the work at the very last of the paper.

**Limitations:**

Although since it is a theoretical paper, I do not think an external ethical review is needed, the limitations, potential societal impacts and ethical concerns, and future work are lacking. Authors may address it later if applicable.

---

> ### Author Rebuttal · Authors · 2023-08-09
>
> Thank you for your positive assessment of our paper. We respond below to your specific concerns but please also check the common response which contains clarifications that were common to all reviewers and will be added to the final version.
>
> - Discussion of Limitations, societal impact and ethical concerns.
>
> We have added a paragraph in the discussion about the limitations of our works, potential future work and connections to more practical applications. Thank you for pointing out these points. Regarding societal impact and ethical concerns, given the nature of the paper -- a theoretical result limiting the extent of Distributional RL -- we do not believe it can have any such impact. The Neurips CFP says “The checklist is filled out in the OpenReview submission form, but you may provide additional information in your paper to support your answers (e.g., a limitations section).” and we believe no additional information is needed. If you have a specific concern about the societal impact of our work that we have overlooked, please let us know and we’ll try to address it.
>
> - More details about experiments.
>
> Thank you for the question. As other reviewers have also raised questions about our experiments, we address this in the common response. First we were able to identify a bug in the computation of the Wasserstein distance in our code and we now have correct Figures (see pdf) that show a much tighter bound. Second, we have explored the tightness of our bound and we are able to make comments on it. We will add this discussion and more details about the experiments in the final version.
>
> - Lack of references.
>
> We are not sure what you are referring to. The subsection “Beyond Expected Rewards” introduces alternative statistical functionals and for CVaR we cite [Rockafellar, 2000]. Our paper has more than 2 pages of references and we are not sure what is missing. Do you have a pointer to relevant papers we may have missed?

---

> > ### Comment · Reviewer_3Rfs · 2023-08-20
> >
> > Thanks the reviewer for the reply. I do not have further new questions.
> > The lacking references problem from my side was mainly about Section 2. But now with re-reading I think I made a mistake about this. Thanks the reviewer for sharp response to correct me.

---

### Author Rebuttal · Authors · 2023-08-09

**We first want to thank all the reviewers** for their constructive feedback and relevant questions that will help clarify the article. There are a few questions raised by several reviewers that we wish to address here.

### Experiments and tightness of our bound
We first want to point out **a minor but impactful bug discovered in the code**, that falsified the experiments of section 3 on the approximation of distribution and statistics. **The corrected results support even more our bounds, and proof method**. Please see the attached pdf for the corrected figures. The method supposed to return the distribution function would return a density function instead. In the notebook code sent in additional material, this can be fixed by changing the line 176 of the 2nd cell, to `probabilities = list(np.cumsum([self.distrib[atom] for atom in atoms]))` instead of `probabilities = [self.distrib[atom] for atom in atoms]`. The experiment was also changed, using a Binomial with $p=0.5$ as parameter, which is more natural.


### Organization of the paper.
A main objective of the paper was to depict a somewhat complete picture of what could be done in Markov Decision Processes with statistics other than expectation of the return. To do so, we wanted to address the 3 main problems that is Policy Evaluation, Control/Planning, and Reinforcement Learning, in that specific order that seemed more natural.
Our central result is Theorem 2 in the Planning section (Section 4), about the Bellman Optimizability. We agree that it is not highlighted enough in the abstract and introduction, which is now corrected.
The section 3’s role is twofold. First, we address the question of Policy Evaluation in the undiscounted setting, and additionally provide error bounds in the case of finite parametrization of the distribution. Doing so, we are led to introduce the notion of Bellman Closedness from the literature, which is relevant to contrast with the upcoming novel definition of Bellman optimizability. We do not see how we could naturally move Section 4 before Section 3.

### Q-Learning in Section 5 is not a contribution
**In Section 5, Algorithm 3 is from [Borkar 2002, 2010], and is not a contribution**. The reason why we report it is to complete the argument of the paper by addressing the learnability of utilities, which is the most important problem in RL. We believe that Q-learning for exponential utilities isn’t well known and we wanted to recall its existence. It is true that we do not talk about its theoretical properties, and we have added more references to point to them. We also emphasized again that this algorithm is not a contribution.

### Definition of the Bellman Optimizability.
The choice of definition was made according to where the idea came from and what it first implied: the impossibility to optimize other statistics, even with distributional reinforcement learning. Hence the definition using a theoretical distributional algorithm. We acknowledge that this definition is cumbersome and makes it harder to understand all the implications of our main result, the theorem 2. We take into account your feedback, and a new definition, more clear and more related to the Bellman Closedness, will be provided in the final version.

---

### Decision · Program_Chairs · 2023-09-21

**Decision:**

Accept (poster)

**Comment:**

The paper is primarily theoretical and it examines a fundamental questions, namely what functionals of rewards can be computed and optimized exactly in an MDP.

The reviewers agree that the paper makes a contribution to a very fundamental RL question, and the paper is well presented and motivated. The empirical evaluation is somewhat limited, but this is not the purpose of the paper.
Overall, the reviewer finds the paper interesting and recommend acceptance.